# Serum leucine-rich α2 glycoprotein as a potential biomarker for systemic inflammation in Parkinson's disease

Takuma Ohmichi[1], Takashi Kasai[1]*, Makiko Shinomoto[1], Fukiko Kitani-Morii[1,2], Yuzo Fujino[1], Kanako Menjo[1], Toshiki Mizuno[1]

1 Department of Neurology, Kyoto Prefectural University of Medicine, Kyoto, Japan, 2 Department of Molecular Pathobiology of Brain Diseases, Kyoto Prefectural University of Medicine, Kyoto, Japan

* kasaita@koto.kpu-m.ac.jp

**Editor:** Giulia Donzuso, University of Catania Department of Surgical and Medical Sciences Advanced Technologies GF Ingrassia: Universita degli Studi di Catania Dipartimento di Scienze Mediche Chirurgiche e Tecnologie Avanzate GF Ingrassia, ITALY

## Abstract

There is ample epidemiological and animal-model evidence suggesting that intestinal inflammation is associated with the development of Parkinson's disease (PD). Leucine-rich α2 glycoprotein (LRG) is a serum inflammatory biomarker used to monitor the activity of autoimmune diseases, including inflammatory bowel diseases. In this study, we aimed to investigate whether serum LRG could be used a biomarker of systemic inflammation in PD and to help distinguish disease states. Serum LRG and C-reactive protein (CRP) levels were measured in 66 patients with PD and 31 age-matched controls. We found that serum LRG levels were statistically significantly higher in the PD group than in the control group (PD: 13.9 ± 4.2 ng/mL, control: 12.1 ± 2.7 ng/mL, p = 0.036). LRG levels were also correlated with Charlson comorbidity index (CCI) and CRP levels. LRG levels in the PD group were correlated with Hoehn and Yahr stages (Spearman's r = 0.40, p = 0.008). LRG levels were statistically significantly elevated in PD patients with dementia as compared to those without dementia (p = 0.0078). Multivariate analysis revealed a statistically significant correlation between PD and serum LRG levels after adjusting for serum CRP levels, and CCI (p = 0.019). We conclude that serum LRG levels could be considered a potential biomarker for systemic inflammation in PD.

## Introduction

Parkinson's disease (PD) is a common neurodegenerative disorder characterized by neuronal α-synuclein aggregation. The underlying mechanisms of neurodegeneration in PD remain unclear. However, a growing body of evidence from *in vitro* and animal models supports the causal role of chronic bowel inflammation in PD pathology [1–5]. This gut-brain connection in PD is supported by the elevated expression of proinflammatory cytokines observed in colon biopsies in patients with PD [6], as well as by an increased risk of PD in patients with inflammatory bowel diseases [7]. Gut-mediated inflammation may be involved in PD development [5], and a slight but statistically significant elevation of systemic inflammatory markers,

**Data Availability Statement:** All relevant data are within the manuscript and its Supporting Information files.

**Funding:** This study was supported by Grants-in-Aid (grant numbers 18K07506 and 21K07466 to TK, grant number 20K16605 to TO) from the Ministry of Education, Culture, Sports, Science and Technology of Japan; and the Takeda Science Foundation grant awarded to TO. The funders had no role in study design, data collection and analysis, decision to publish, or preparation of the manuscript.

**Competing interests:** TO received Grants-in-Aid from the Japan Society for the Promotion of Science (JSPS KAKENHI) (grant number JP20K16605) and a grant from the Takeda Science Foundation. TK received JSPS KAKENHI (grant numbers JP21K07466, JP21K07855). FKM received JSPS KAKENHI (grant number JP19K16924) and a Japan heart foundation research grant. TM received JSPS KAKENHI (grant number JP18K07533), a Grant-in-Aid for Research on Intractable Disease (21FC0201, 21FC1002) from the Japanese Ministry of Health, Labor, and Welfare, Japan, and AMED (grant numbers JP21dm0307003s0105, JP21ek0109516s0301, JP21ek0210120s0202). MS, YF, and KM have no financial disclosure to declare. This does not alter our adherence to PLOS ONE policies on sharing data and materials.

including serum C-reactive protein (CRP), is well established in PD and is thought to reflect gut mucosal inflammation [8]. Additionally, some systematic inflammatory markers are reported to be associated with clinical outcomes in regard motor and cognitive performance as well as cerebrospinal fluid (CSF) levels of PD-specific biomarkers [9].

Leucine-rich α2 glycoprotein (LRG), a 50-kDa glycoprotein with leucine-rich repeats, acts as a serum biomarker. This glycoprotein was detected using proteomics in patients with rheumatoid arthritis [10]. Serum LRG levels have been reported to be elevated in various autoimmune diseases, correlating with disease activity in rheumatoid arthritis [11], systemic lupus erythematosus [12], adult-onset Still's disease [13], and systemic juvenile idiopathic arthritis [14]. The upregulation of LRG is associated with interleukin (IL) 6, IL-1β, tumor necrosis factor-α (TNFα), and IL-22 levels, among other biomarkers, and this protein is produced in inflamed organs and in the liver [15]. Additionally, previous studies have reported that compared to CRP levels, serum LRG levels have a stronger correlation with disease activity in ulcerative colitis [15–17], and that serum LRG is clinically used for disease activity monitoring in inflamatory bowel diseases (IBD) [18]. One previous report showed that CSF levels of LRG were elevated in patients with PD with dementia (PDD) and dementia with Lewy bodies (DLB) as compared to controls [19]. However, to our knowledge, the value of serum LRG levels has not been thoroughly investigated in PD patients. Therefore, we conducted a single-center, case-control study to compare serum LRG and CRP concentrations between patients with PD and controls. Multivariate analysis revealed a statistically significant correlation between PD and serum LRG levels, demonstrating that serum LRG levels could be considered a potential biomarker for systemic inflammation in PD.

## Materials and methods

All study participants or their nearest living relative provided written informed consent before participation. Moreover, the study protocols were approved by the University Ethics Committee (ERB-G-12, Kyoto Prefectural University of Medicine, Kyoto, Japan), and the study was designed and performed in accordance with the principles of the Declaration of Helsinki.

We enrolled patients diagnosed with clinically probable PD based on Movement Disorder Society (MDS) criteria. Additionally, age-matched unrelated individuals without parkinsonism, cognitive impairment, or autonomic neuropathy were enrolled as controls. Subjects were excluded if they had inflammatory diseases at the time of blood collection, such as collagen disorders, autoimmune diseases, or incidental infectious diseases, including urinary tract infection, bronchitis, aspiration pneumonia, or fever of unknown origin.

All participants visited the Kyoto Prefectural University of Medicine between October 2017 and August 2020. Blood samples were collected from all patients in tubes with a clot activator and gel separator (Terumo, Tokyo, Japan). After collection, the serum was separated by centrifugation for 10 min at 2,000 × g and stored at -80°C until analysis. Serum LRG levels were measured by Nanopia LRG according to a previously established method using latex agglutination (Sekisui Medical Co Ltd, Tokyo, Japan). Serum CRP levels were examined using a routine analyzer (LABOSPECT 008-α, Hitachi High-Technologies, Tokyo, Japan).

To estimate the impact of comorbid diseases on systemic inflammation, the Charlson comorbidity index (CCI) [20] was evaluated from clinical records in both the PD and control groups. All participants in the PD group were assessed according to Hoehn and Yahr (HY) stage, duration from disease onset, and levodopa equivalent daily dose (LEDD) at sample collection. Presence of dementia was determined based on the clinician's assessment.

The present study analyzed the early heart/mediastinum (H/M) ratio and mean specific binding ratio of the bilateral striatum (SBR) in patients who underwent myocardial imaging

with [123]I-metaiodobenzylguanidine (MIBG) and dopamine transporter (DAT) scans using [123]I-Ioflupane within three years from sample collection. MDS-Unified Parkinson's Disease Rating Scale (UPDRS) part I, II and III scores were assessed within one year of sample collection [21]. For patients whose UPDRS part III scores were not assessed within this period, the authors requested the treating physicians to estimate UPDRS part III scores at sample collection based on the UPDRS part 3 score collected more than one year prior and/or after sample collection, as well as the patients' own description of motor symptoms at sample collection, as described in clinical records. The actual measured UPDRS part III score and the abovementioned estimated UPDRS part III score were integrated and analyzed.

A constipation grade was likewise developed in this study to classify the severity of constipation in the PD group based on laxative use for constipation at sample collection. Patients were classified as having grade 0, 1, 2, or 3 constipation, defined as "no medication for constipation control," "regular use of a single type of laxative," "regular use of two or more types of laxatives," and "regular use of enema in addition to laxatives," respectively.

Comparisons between independent groups were performed using the Mann-Whitney U test. Fisher's exact test was used to evaluate the statistical significance of the categorical variables. Correlational analysis was conducted using Spearman's rank correlation test. The above analyses were carried out using GraphPad Prism software version 6.0 (San Diego, CA, USA).

Associations between biomarker and clinical characteristics were analyzed using multivariate regression models. In the multiple regression analysis evaluating the contribution of biomarker values and the impact of comorbid diseases in regard to the presence or absence of PD, we used a binominal logistic regression model with the presence or absence of PD considered as the dependent variable and LRG levels, CRP levels, and CCI scores taken as the explanatory variables. Goodness of fit and overall model evaluation were assessed using the Hosmer and Lemeshow and likelihood ratio tests, respectively. In multiple regression analyses regarding correlations between the severity of motor dysfunction (i.e., HY stage and MDS-UPDRS part III score), biomarker values, and comorbid diseases, we used multiple linear regression models with HY stage and MDS-UPDRS part III score as the dependent variables, and LRG levels, CRP levels, and CCI scores taken as explanatory variables. F-tests were used to assess how each multivariate linear regression model fit the data.

The level of statistical significance was set at $p < 0.05$. JMP 12.2.0 statistical software (SAS Institute Inc., Cary, NC, USA) was used for multivariate analyses. Furthermore, as a subgroup analysis, patients with PD and controls were matched at a ratio of 1:1 using the propensity score matching approach. Matching factors included age, CCI scores, and serum C-reactive protein (CRP) levels.

## Results

A total of 66 patients with PD and 31 controls were registered in the PD and control groups, respectively. The control group comprised 26 healthy subjects, three patients with cervical spondylosis, one patient with vocal spasms, and one patient with migraines. The demographic data of the two groups are shown in Table 1 and S1 Table. There were no statistically significant differences in age, sex, or CCI score between the two groups (p = 0.190, p = 0.459, and p = 0.521, respectively).

The serum levels of LRG were statistically significantly higher in the PD group than in the control group (13.9 ± 4.2 μg/mL vs. 12.1 ± 2.7 μg/mL, p = 0.036) (Fig 1A). We found that the statistically significant difference in serum LRG levels between the PD and control group was preserved after exclusion of an outlier case (LRG level: 35.3 ng/mL) in the PD group (p = 0.045). The area under the receiver operating characteristic curve was 0.63 (Fig 1B).

**Table 1. Patient demographic and medical characteristics.**

| | PD group (n = 66) | Control group (n = 31) | p-value |
|---|---|---|---|
| Age (years) | 72.5 (41–89) | 62 (36–86) | 0.19 |
| Sex (number of female patients) | 26 [39%] | 13 [42%] | 0.82 |
| Presence of dementia (number of patients with dementia) | 15 [23%] | - | |
| CCI | 0 (0–6) | 0 (0–5) | 0.52 |
| HY stage | | | |
| 0 | 2 [3%] | - | - |
| 1 | 6 [9%] | - | - |
| 2 | 8 [12%] | - | - |
| 3 | 32 [48%] | - | - |
| 4 | 16 [24%] | - | - |
| 5 | 2 [3%] | - | - |
| MDS-UPDRS-I (points) * | 10 (3–27) | - | - |
| MDS-UPDRS-II (points) * | 18 (5–46) | - | - |
| MDS-UPDRS-III (points) ** | 21 (2–56) | | |
| LEDD (mg/day) | 275 (0–1331.25) | - | - |
| Duration from onset (months) | 28 (1–367) | - | - |
| Constipation scale | 0 (0–3) | | |
| H/M ratio of MIBG (early phase) *** | 1.96 (0.96–3.63) | | |
| SBR on DAT imaging **** | 3.20 (0.27–6.45) | - | - |
| Serum CRP level (ng/mL) | 0.05 (0–0.94) | 0.04 (0–0.71) | 0.60 |
| Serum LRG level (ng/mL) | 12.9 (7.8–35.3) | 11.6 (8.9–18.7) | 0.036 |

Abbreviations: CCI, Charlson comorbidity index; DAT, dopamine transporter; H/M, heart/mediastinum; HY, Hoehn & Yahr; LEDD, levodopa equivalent daily dose; MDS-UPDRS, Movement Disorder Society-Unified Parkinson's Disease Rating Scale; MIBG, myocardial imaging with [123]I-metaiodobenzylguanidine; PD, Parkinson's disease; SBR, specific binding ratio.

Continuous variables are expressed as medians (ranges). The percentages of subjects in each group are presented using square brackets.

* 28 participants were tested.

** MDS-UPDRS part III were obtained from a total of 63 participants. MDS-UPDRS part III scores of 31 participants were actually measured within one year before and/or after sample collection, while those of the other 32 participants were estimated by their physician based on the patients' clinical records.

*** 21 participants were tested.

**** 47 participants were tested.

Moreover, when using Youden's index to examine the cutoff value for LRG concentrations, a cutoff value of 13.20 yielded a sensitivity of 0.7742 and a specificity of 0.4848.

In contrast, the serum levels of CRP did not differ statistically significantly between the two groups (PD: 0.12 ± 0.15 µg/mL, control: 0.12 ± 0.18 µg/mL) (Fig 1C). Furthermore, the LRG serum levels in the PD group were weakly correlated with HY stage (Spearman r = 0.40, p = 0.008) (Fig 2A) as well as with UPDRS part III scores (Spearman r = 0.34, p = 0.0006) (S2C Fig), but not with LEDD, constipation grade, or duration from onset (Fig 2C–2E). In a sub-analysis of the PD group patients who were assessed in regard to their UPDRS part I and part II scores, the H/M ratio in the early phase on MIBG myocardial imaging, or SBR on DAT imaging, we found no statistically significant correlation between those parameters and LRG levels (S2A–S2D Fig). Serum LRG levels were not correlated with age in either group (Fig 2F and 2G). When examining LRG levels with respect to the presence or absence of dementia, we found that LRG levels were statistically significantly elevated in PD patients with dementia as compared to those without dementia (p = 0.0078) (Fig 2H).

To investigate the correlations of age, CRP levels, and the burden of comorbid diseases with LRG levels, we univariately analyzed these variables after integrating the PD and control

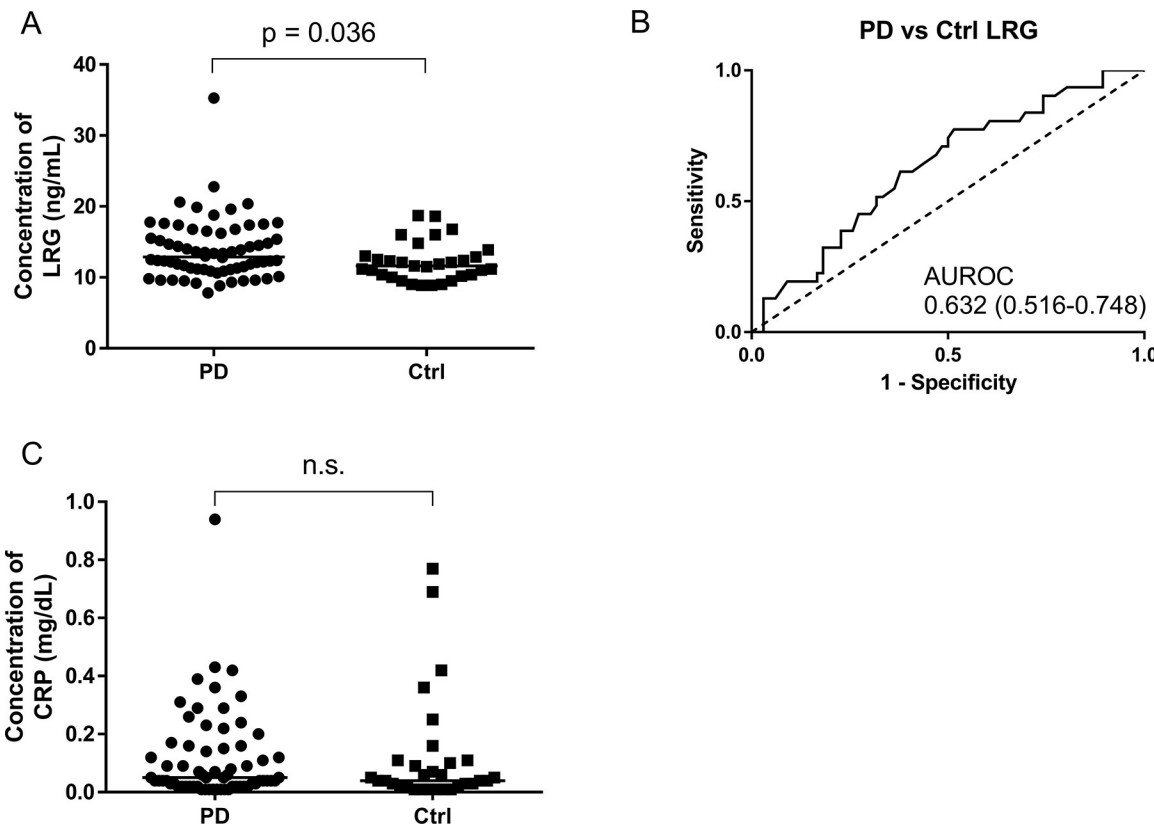

**Fig 1. Serum LRG and CRP levels in the PD and control groups.** (A) Scatter plot of LRG levels. (B) ROC curve analysis for diagnosis of patients with Parkinson's disease (PD) and controls (Ctrl) by LRG levels. (C) Scatter plot of CRP levels. Bars indicate medians. The p-values generated by the Mann–Whitney U test are shown here. Abbreviations: AUROC, area under the receiver operating characteristic curve; Ctrl, controls; CRP, C-reactive protein; LRG, leucine-rich α2 glycoprotein; n.s.: not significant; PD, Parkinson's disease; ROC, receiver operating characteristics.

groups. As expected from the above results, serum LRG levels were correlated with PD. Furthermore, a strong positive correlation between serum LRG and CRP levels was observed. LRG levels were also statistically significantly correlated with CCI (Table 2).

A binominal logistic regression analysis with PD as the dependent variable and LRG, serum levels of CRP, and CCI scores as the explanatory variables was conducted to estimate the relationship between these three factors and the presence of PD. There was a significant effect of LRG levels on the presence of PD (p = 0.019), while those of CRP levels or CCI were not significant (Table 3). PD and control groups were matched using the propensity score approach, and serum LRG levels were statistically significantly higher in the PD group than in the control group (S2 Table and S1 Fig).

The correlations between LRG levels and motor function scores (i.e., HY stages and MDS-UPDRS part III scores) were also analyzed using multiple linear regression models with LRG levels as the dependent variables, and CRP levels, CCI scores, and HY stage or MDS-UPDRS part III scores as the explanatory variables. After adjusted CCI and CRP levels, HY stages as well as MDS-UPDRS part III scores were statistically significantly correlated with LRG levels (β = 0.262, p = 0.026 and β = 0.285, p = 0.010, respectively) (Table 4A and 4B). For additional analyses related to the relationship between LRG values and the presence of dementia in the PD group, we conducted a binominal logistic regression analysis with the presence of dementia as the dependent variable and LRG, serum levels of CRP, and CCI scores as the

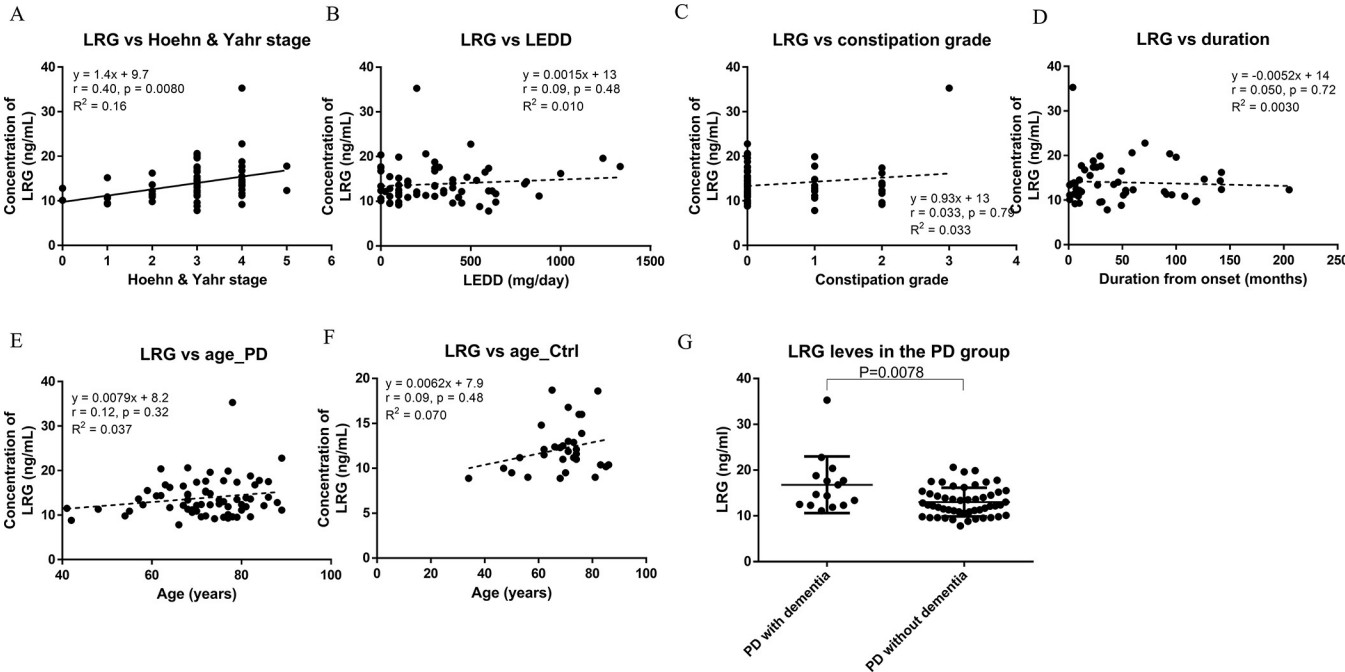

**Fig 2. Serum LRG levels and clinical data in the enrolled participants.** Associations between serum levels of LRG and Hoehn & Yahr stage (A), LEDD (B), constipation grade (C), duration from onset (D), age in the PD group (E), and age in the control group (F). A comparison of serum LRG levels between PD patients with and without dementia is shown in (G). Abbreviations: DAT, dopamine transporter; LEDD, levodopa equivalent daily dose; PD, Parkinson's disease; SBR, specific binding ratio of the bilateral striatum; UPDRS, Unified Parkinson's Disease Rating Scale.

explanatory variables. There was a significant effect of LRG levels on the presence of dementia (p = 0.019), while those of CRP levels or CCI were not significant (Table 4C).

## Discussion

This study demonstrated a statistically significant elevation in serum LRG levels in patients with PD as compared to age-matched controls, and LRG elevation was statistically significantly more prominent in PD patients with dementia than in those without dementia. Moreover, consistent with the ability of proinflammatory cytokines to induce LRG, we found that LRG levels were correlated with CRP levels and CCI scores, which are reported to be higher in patients with PD than in controls [8]. However, the presence of PD remained independently associated with serum LRG levels after adjusting for age, CCI scores, and CRP levels. As the difference in serum CRP levels between the PD and control groups did not reach statistical

**Table 2. Correlations between serum LRG levels and clinical characteristics.**

|  | Correlation coefficient | p-value |
|---|---|---|
| Age | 0.155 | 0.130 |
| Presence of PD | 0.214 | 0.0356 |
| CCI | 0.254 | 0.0120 |
| Serum CRP levels | 0.406 | <0.0001 |

Abbreviations: CCI, Charlson comorbidity index; CRP, C-reactive protein; PD, Parkinson's disease.

Univariate analysis of serum LRG for each clinical characteristic. The explanatory variables include age, the presence of PD (presence of PD: 1; absence of PD: 0), CCI, and serum CRP levels.

**Table 3. Logistic regression analysis for the presence of Parkinson's disease.**

|  | B value | SE | $\chi^2$ value | p-value | OR |
|---|---|---|---|---|---|
| Serum LRG | 0.206 | 0.088 | 5.53 | 0.019 | 1.229 |
| Serum CRP | -1.342 | 1.447 | 0.86 | 0.354 | 0.261 |
| CCI | -0.246 | 0.180 | 1.87 | 0.171 | 0.215 |

Abbreviations: CCI, Charlson comorbidity index; CRP, C-reactive protein; df, degree of freedom; PD, Parkinson's disease; SE, standard error.

The dependent variable set as the presence of PD (presence of PD: 1; absence of PD: 0). The explanatory variables include serum LRG, CRP levels and CCI scores.

Hosmer and Lemeshow Goodness of Fit test: $\chi^2 = 113.7$, df = 3, p-value = 0.054.

Overall model evaluation using the likelihood ratio test: $\chi^2 = 7.8$, df = 3, p-value = 0.0498.

significance in the present study, we conclude that serum LRG levels might represent a better diagnostic biomarker of PD than serum CRP levels. This observation was in accordance with previous research findings that serum LRG levels were elevated in a subpopulation of patients with Crohn's disease presenting with normal CRP levels [16], and were associated with disease activity in patients with ulcerative colitis presenting with normal CRP levels [15].

To the best of our knowledge, this is the first report showing the significant diagnostic value of serum LRG for PD diagnosis. There was one previous report measuring serum LRG levels in various diseases, wherein no statistically significant differences were observed between 21 patients with neurological disorders, including 13 patients with PD, and 21 controls [22]. The

**Table 4. Regression analysis for evaluating motor function scores and the presence of dementia.**

(A) Multiple linear regression of LRG levels against HY stage, CCI and CRP in the PD group

|  | β coefficient | t value | p-value | F statistics | | R squared | Adjusted R squared | Variance inflation factor |
|---|---|---|---|---|---|---|---|---|
|  |  |  |  | F value | p-value |  |  |  |
| HY stages | 0.262 | 2.28 | 0.026 | 9.84 | < 0.001 | 0.32 | 0.28 | 1.21 |
| CCI | 0.002 | 0.02 | 0.98 |  |  |  |  | 1.37 |
| Serum CRP levels | 0.453 | 3.99 | <0.001 |  |  |  |  | 1.18 |

(B) Multiple linear regression of LRG levels against UPDRS part III scores, CCI and CRP in the PD group

|  | β coefficient | t value | p-value | F statistics | | R squared | Adjusted R squared | Variance inflation factor |
|---|---|---|---|---|---|---|---|---|
|  |  |  |  | F value | p-value |  |  |  |
| UPDRS part III scores | 0.285 | 2.66 | 0.010 | 11.48 | <0.001 | 0.39 | 0.34 | 1.28 |
| CCI | 0.030 | 0.25 | 0.802 |  |  |  |  | 1.20 |
| Serum CRP levels | 0.493 | 4.34 | <0.001 |  |  |  |  | 1.07 |

(C) Logistic regression analysis of the presence of dementia against each variable in the PD group

|  | B value | SE | $\chi^2$ value | p-value | OR |
|---|---|---|---|---|---|
| Serum LRG | 0.250 | 0.106 | 5.52 | 0.019 | 1.284 |
| Serum CRP | -1.972 | 2.647 | 0.56 | 0.456 | 0.139 |
| CCI | 0.288 | 0.251 | 1.32 | 0.251 | 1.334 |

(A) The dependent variable set as LRG levels. The explanatory variables include HY stages, CRP levels and CCI scores.

(B) The dependent variable set as LRG levels. The explanatory variables include UPDRS part III scores, CRP levels and CCI scores.

(C) The dependent variable set as the presence of dementia (presence of dementia: 1; absence of dementia: 0). The explanatory variables include serum LRG, CRP levels and CCI scores.

Hosmer and Lemeshow Goodness of Fit test: $\chi^2 = 8.210$, df = 7, p-value = 0.314.

Overall model evaluation using the likelihood ratio test: $\chi^2 = 10.245$, df = 3, p-value = 0.017.

Abbreviations: CCI, Charlson comorbidity index; CRP, C-reactive protein; LRG, leucine-rich α2 glycoprotein; MDS-UPDRS, Movement Disorder Society-Unified Parkinson's Disease Rating Scale; PD, Parkinson's disease.

inconsistency between our results and those of this previous study may be due to differences in the number of samples and the homogeneity of the evaluated disease group across investigations. Regarding CSF findings in neurological diseases, Li et al. initially found an elevated LRG level in the CSF to be a marker for idiopathic normal pressure hydrocephalus based on proteomic analysis [23]. Their subsequent study also showed that CSF LRG levels could discriminate patients with PDD and DLB from controls with high accuracy (sensitivity = 0.929, specificity = 0.964) [19]. Although no evidence regarding the relationship between CSF and serum LRG levels was found, their results are consistent with our findings, including the finding that LRG is more elevated in PD patients with dementia than in those without dementia.

In this study, there were no statistically significant correlations between serum LRG levels and age, MDS-UPDRS part I or II scores, LEDD, duration from onset, constipation grade, H/M ratio in the MIBG myocardial images, or SBR on DAT imaging, whereas we observed a weak correlation between serum LRG levels and HY stage. Taken together with the fact that MDS-UPDRS part III scores were also correlated to serum LRG level, serum LRG levels might be associated with motor dysfunction in PD though some of our UPDRS part III scores were imputed by treating physicians on the basis of the symptoms reported by the patients rather than with an actual examination and therefore should be interpreted with caution. We note that inconsistent results have been reported regarding the relationship between HY stage and serum inflammatory markers. For example, some studies have reported a positive relationship between HY stage and serum levels of TNFα [24] and CRP [25, 26], while others have not reported a similar association [27, 28]. Future validation of the positive correlation between serum LRG and HY stages is required. Since LRG expression was markedly increased in a dextran sodium sulfate-induced enteritis model and serum LRG levels were likewise elevated, reflecting increased IBD activity [29], we expected that LRG would correlate with gastrointestinal symptoms of PD. However, we found no correlation between LRG and constipation grade. This may be because intestinal inflammation in PD patients with constipation was not as high as we expected, and the small number of cases resulted in insufficient statistical power.

Regarding a potential relationship between SBR on DAT imaging and inflammatory biomarkers, although we did not find such a correlation between SBR on DAT and LRG in the current study, there is a report that SBR on DAT imaging as well as the neutrophil to lymphocyte ratio were correlated only in tremor dominant PD [30]. This inconsistency may be due to the fact that we excluded participants with inflammatory disease as well as that the sample size in the PD group in the current study was too small to conduct sub-analyses based on clinical subtypes.

Although elevated serum CRP levels in patients with PD as compared to controls have been confirmed in a prior meta-analysis [6], this study did not reproduce these previously reported results. This may have been due to the exclusion of participants with inflammatory diseases. We also acknowledge that the relatively small enrolled sample size was a major limitation of the study, which might have weakened the statistical power for detecting an association between CRP levels and the presence of PD. Another important limitation is that this study was a retrospective analysis. In fact, UPDRS part I and II scores and MIBG were performed in less than half of the enrolled subjects, which may have led to lower statistical power for the correlation analysis evaluating the association between these findings and serum LRG levels. We did not assess constipation, an important non-motor symptom of PD related to intestinal inflammation, by established examinations (such as the Scale for Outcomes in Parkinson's disease-Autonomic [SCOPA-AUT] questionnaire or the Non-Motor Symptoms Scale [NMSS]), and only evaluated it by a non-validated scale obtained from a retrospective review of patient medical records. This limitation may have prevented us from detecting an association between gastrointestinal symptoms and LRG levels.

## Conclusion

We found that serum LRG levels were statistically significantly elevated in patients with PD as compared in controls. Given that the difference in serum CRP levels between the PD and control groups did not reach statistical significance in the present study, we conclude that serum LRG levels might represent a better biomarker for the systemic inflammation seen in PD as compared with CRP. However, considering the small sample size of this study, the present results require validation in larger cohorts.

## Supporting information

**S1 Table.** Clinical data of participants in the Parkinson's disease (a) and control groups (b). Abbreviations: CCI, Charlson comorbidity index; CRP, C-reactive protein; DAT, dopamine transporter single photon emission computed tomography; HY, Hoehn & Yahr; LEDD, levodopa equivalent daily dose; LRG, leucine-rich alpha-2 glycoprotein; MIBG H/M, early heart/ mediastinum ratio of myocardial imaging with 123I-metaiodobenzylguanidine; N/A, not available; SBR, specific binding ratio.
(XLSX)

**S2 Table. Patient demographics following propensity score matching.**
(DOCX)

**S1 Fig. Serum LRG levels in the PD and control groups following propensity score matching.** Evaluations in 30 pairs of PD and control groups matched using a propensity score matching approach. (A) Scatter plot of LRG levels. Serum levels of LRG were statistically significantly higher in the PD group than in the control group (14.6 ± 4.9 μg/mL vs 12.3 ± 2.7 μg/ mL, p = 0.019). P-values generated by the Mann–Whitney U test are shown here. (B) Receiver operating characteristic curve analysis for diagnosis of PD by LRG levels. Abbreviations: AUROC, area under the receiver operating characteristics curve; Ctrl, controls; LRG, leucine-rich α2 glycoprotein; PD, Parkinson's disease.
(PPTX)

**S2 Fig. Serum LRG levels and clinical data in the enrolled participants.** (A) MDS-UPDRS part I scores. (B) MDS-UPDRS part II scores. (C) MDS-UPDRS part III scores. (D) H/M ratio in the early phase on MIBG myocardial images. (E) SBR on DAT imaging. There were no statistically significant correlations between these parameters and serum LRG levels. Abbreviations: DAT, dopamine transporter single photon emission computed tomography; H/M, heart/mediastinum; LEDD, levodopa equivalent daily dose; LRG, leucine-rich α2 glycoprotein; MIBG, myocardial imaging with [123]I-metaiodobenzylguanidine; MDS, Movement Disorder Society; UPDRS, Unified Parkinson's Disease Rating Scale; SBR, specific binding ratio.
(PPTX)

## Acknowledgments

We would like to thank Editage (www.editage.com) for English language editing.

## Author Contributions

**Conceptualization:** Takuma Ohmichi, Takashi Kasai.

**Data curation:** Takuma Ohmichi, Takashi Kasai, Makiko Shinomoto, Fukiko Kitani-Morii, Yuzo Fujino, Kanako Menjo.

**Formal analysis:** Takuma Ohmichi, Takashi Kasai, Makiko Shinomoto, Fukiko Kitani-Morii.

**Funding acquisition:** Takuma Ohmichi, Takashi Kasai.

**Investigation:** Takuma Ohmichi, Takashi Kasai.

**Methodology:** Takuma Ohmichi, Takashi Kasai.

**Project administration:** Takuma Ohmichi, Takashi Kasai.

**Supervision:** Toshiki Mizuno.

**Writing – original draft:** Takuma Ohmichi.

**Writing – review & editing:** Takuma Ohmichi, Takashi Kasai, Makiko Shinomoto, Fukiko Kitani-Morii, Yuzo Fujino, Kanako Menjo, Toshiki Mizuno.

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
