## [Decision Letter · Decision Letter 0]

25 Nov 2022

PONE-D-22-21867Serum leucine-rich α2 glycoprotein as a potential biomarker for the systemic inflammation in Parkinson's disease.PLOS ONE

Dear Dr. Takashi Kasai

Thank you for submitting your manuscript to PLOS ONE. After careful consideration, we feel that it has merit but does not fully meet PLOS ONE’s publication criteria as it currently stands. Therefore, we invite you to submit a revised version of the manuscript that addresses the points raised during the review process.

We look forward to receiving your revised manuscript.

Kind regards,

Giulia Donzuso

Academic Editor

PLOS ONE

Journal Requirements:

This study was supported by Grants-in-Aid (grant numbers 18K07506 and 21K07466 to TK, grant number 20K16605 to TO) from the Ministry of Education, Culture, Sports, Science and Technology of Japan; and the Takeda Science Foundation grant awarded to TO. 

However, funding information should not appear in the Acknowledgments section or other areas of your manuscript. We will only publish funding information present in the Funding Statement section of the online submission form. 

This study was supported by Grants-in-Aid (grant numbers 18K07506 and 21K07466 to TK, grant number 20K16605 to TO) from the Ministry of Education, Culture, Sports, Science and Technology of Japan; and the Takeda Science Foundation grant awarded to TO. The funders had no role in study design, data collection and analysis, decision to publish, or preparation of the manuscript.

TO received Grants-in-Aid from the Japan Society for the Promotion of Science (JSPS KAKENHI) (grant number JP20K16605) and a grant from the Takeda Science Foundation. TK received JSPS KAKENHI (grant numbers JP21K07466, JP21K07855). FKM received JSPS KAKENHI (grant number JP19K16924) and a Japan heart foundation research grant. TM received JSPS KAKENHI (grant number JP18K07533), a Grant-in-Aid for Research on Intractable Disease (21FC0201, 21FC1002) from the Japanese Ministry of Health, Labor, and Welfare, Japan, and AMED (grant numbers JP21dm0307003s0105, JP21ek0109516s0301, JP21ek0210120s0202). MS, YF, and KM have no financial disclosure to declare. 

Reviewers' comments:

Reviewer's Responses to Questions

**Comments to the Author**

1. Is the manuscript technically sound, and do the data support the conclusions?

Reviewer #1: Yes

Reviewer #2: Partly

2. Has the statistical analysis been performed appropriately and rigorously? 

Reviewer #1: Yes

Reviewer #2: No

3. Have the authors made all data underlying the findings in their manuscript fully available?

Reviewer #1: Yes

Reviewer #2: Yes

4. Is the manuscript presented in an intelligible fashion and written in standard English?

Reviewer #1: Yes

Reviewer #2: Yes

5. Review Comments to the Author

Reviewer #1: Reviewer’s notes on manuscript Serum leucine-rich α2 glycoprotein as a potential biomarker for the systemic inflammation in Parkinson's disease.

In this paper the authors compared the levels of an inflammatory biomarker (leuchine-rich α2 glycoprotein , LRG) in a sample of PD patients with a group of healthy controls and found that LRG was higher in PD even after correction by known confounders. The article has its own relevant message, especially considering that inflammation has a major role in PD development and progression and the definition of a new biomarker could help in further understanding of the consequences of inflammation in PD. In my opinion the article could be published after major revisions, considering the lack of some relevant clinical data of PD patients and an overall improvement of the discussion.

Below my specific comments.

Introduction

Please, add a reference in page 3, line 52 when enumerating the conditions where a higher level of LRG has been found.

Methods

Do the authors mean within three years from the sample collection when they write “within three years from imaging” (page 5, line 84)?

There are several validated scale that have specific items for constipation such as the SCOPA-AUT or the Non Motor Symptoms Scale (NMSS) extensively used in PD, why the authors chose a non-validated scale?

In my opinion, one of the major drawbacks of this research is the lack of correlation of LRG with the UPDRS-III, since it represents the most reliable indicator of motor impairment in PD patients, regardless the general level of disability measured by the HY scale.

Why did some patients underwent the execution of MIBG imaging? Was there an initial suspicion of atypical parkinsonisms? Please, detail how the data have been acquired (i.e. standard examination of parkinsonism patients?), and if there are significant differences between the two subgroups (MIBG and DATSCAN) and the general sample.

Results

The number of missing values on UPDRS part I and II is not acceptable, since it represents almost half of the entire sample, as such analysis involving these two variables might not be relevant for the significance of the research, also explaining the lack of correlation of LRG levels with both Part I and Part II. Moreover, the weak but significant correlation with HY stage could have been more extensively explored with UPDRS III that unfortunately is lacking. Please, add the UPDRS part III score and correlate it with LRG. Moreover I would like to suggest considering the analysis of UPDRS part I and II as subgroup analysis, since its scientific relevance is at least dubious due to the reduced sample size.

Is there a reason why the authors decided to correlate with the constipation grade? Even if not significant, the need for this analysis should be better explained in the discussion section.

In my opinion, a sensitivity analysis should be conducted after excluding a PD patient that has an outlier value (35.3 ng/ml, as seen from Supplementary Table 1 and Figure 1), this could help strengthen the results of the study.

Discussion

I think the authors should expand a little the discussion on the lack of association with DATSCAN or MIBG as to interpret this data in relation with inflammation in PD, since previous reports have found significant associations between DATSCAN and inflammation even if only in TD subtype ( see Sanjari Moghaddam H, Ghazi Sherbaf F, Mojtahed Zadeh M, Ashraf-Ganjouei A, Aarabi MH. Association Between Peripheral Inflammation and DATSCAN Data of the Striatal Nuclei in Different Motor Subtypes of Parkinson Disease. Front Neurol. 2018 Apr 16;9:234. doi: 10.3389/fneur.2018.00234. PMID: 29713303; PMCID: PMC5911462.)

Since previous studies have showed an association with PDD, it could be interesting to suggest that further studies should explore the LRG levels in PD patients with MCI/PDD vs normal cognition.

Reviewer #2: In the present study the Authors aimed to assess the potential role of serum LRG as a biomarker of systemic inflammation in PD, finding significantly increased LRG levels in PD in respect with controls as well as a positive correlation between LRG levels and HY stage in PD group. The topic is interesting; the study design is appropriate; the paper is well written. Nevertheless, I have some major points of criticism.

A) In this study, HY stage was used to evaluate motor impairment in PD. MDS UPDRS-III would be also useful in order to provide a more accurate motor evaluation.

B) Significantly higher serum LRG levels were found in PD as compared with controls. No significant differences were found in age, sex, CCI and CRP levels between study groups. Nevertheless, considering the whole study population, a significant independent positive correlation was found between LRG and CRP serum levels, CCI and presence of PD. Therefore, I believe that a multivariate logistic regression analysis considering PD as a dependent variable is needed in order to assess the possible confounding role of these variables in the association found between PD and higher LRG levels.

Similarly, multiple linear regression analysis should be performed to confirm the positive correlation between serum LRG levels and HY stage in PD group after adjusting at least by CCI and CRP levels. All of these findings should be reported in tables.

C) ROC analysis was performed with a reported AUC of 0.63 considering LRG levels to differentiate the two study group. A possible cut-off value with its best sensibility and specificity in distinguish PD from controls should be indicated.

D) In line 162 the Authors state that “this is the first report showing the value of serum LRG for PD diagnosis”. Nevertheless, other Authors previously assessed serum LRG levels in PD, showing no differences in respect with controls (see Weivoda S, Andersen JD, Skogen A, Schlievert PM, Fontana D, Schacker T, Tuite P, Dubinsky JM, Jemmerson R. ELISA for human serum leucine-rich alpha-2-glycoprotein-1 employing cytochrome c as the capturing ligand. J Immunol Methods. 2008 Jul 20;336(1):22-9. doi: 10.1016/j.jim.2008.03.004. Epub 2008 Apr 4. PMID: 18436231; PMCID: PMC7094546). This point should be discussed.

Minor comments:

A) Serum LRG and CRP levels should be showed in Table 1, which could be renamed “Clinical and demographic characteristics”.

---

## [Author Response · Author response to Decision Letter 0]

29 Dec 2022

Thank you for your careful peer review. Please find all our comments in the Responce to Reviewers file.

---

## [Decision Letter · Decision Letter 1]

17 Jan 2023

PONE-D-22-21867R1Serum leucine-rich α2 glycoprotein as a potential biomarker for the systemic inflammation in Parkinson's disease.PLOS ONE

Dear Dr. Kasai

Thank you for submitting your manuscript to PLOS ONE. After careful consideration, we feel that it has merit but does not fully meet PLOS ONE’s publication criteria as it currently stands. Therefore, we invite you to submit a revised version of the manuscript that addresses the points raised during the review process.

We look forward to receiving your revised manuscript.

Kind regards,

Giulia Donzuso

Academic Editor

PLOS ONE

Journal Requirements:

Additional Editor Comments:

Dear Authors,

your revisions improved the quality of the manuscript. However, the reviewers needs some minor adjustments before publication.

Reviewers' comments:

Reviewer's Responses to Questions

**Comments to the Author**

1. If the authors have adequately addressed your comments raised in a previous round of review and you feel that this manuscript is now acceptable for publication, you may indicate that here to bypass the “Comments to the Author” section, enter your conflict of interest statement in the “Confidential to Editor” section, and submit your "Accept" recommendation.

Reviewer #1: All comments have been addressed

Reviewer #2: All comments have been addressed

2. Is the manuscript technically sound, and do the data support the conclusions?

Reviewer #1: Yes

Reviewer #2: Partly

3. Has the statistical analysis been performed appropriately and rigorously? 

Reviewer #1: Yes

Reviewer #2: No

4. Have the authors made all data underlying the findings in their manuscript fully available?

Reviewer #1: Yes

Reviewer #2: Yes

5. Is the manuscript presented in an intelligible fashion and written in standard English?

Reviewer #1: Yes

Reviewer #2: Yes

6. Review Comments to the Author

Reviewer #1: The authors have extensively solved the many queries suggested, improving the quality of the manuscript. However, the manuscript needs some minor adjustments before publication:

- Add a limitation concerning the UPDRS III analysis in the discussion, stressing out that the association with UPDRS III should be considered with caution because almost the 25% of UPDRS III data has been imputed by treating physicians on the basis of the symptoms reported by the patients, rather than with an actual examination. This represents a differential bias that could impact the results of the study.

- I find the results of the analysis on dementia interesting, but the authors should add few lines in the methods section on how dementia was diagnosed in these patients (i.e. using MMSE, MOCA or just by clinician assessment?)

I appreciate the efforts of the authors in improving the English of the manuscript, however several typos can still be found and there are other minor corrections such as:

- Page 7, lines 76-79 reporting an entire paragraph that I think does not belong there (typo?)

- Tremor dominant and non tremoristic (page 19, line 301)

- Please be consistent throughout the text for the UPDRS part III (i.e. always use “part III” or “part 3”, not both in the text and figures)

Reviewer #2: I have some major points of criticism:

A) MDS UPDRS-III score were retrospectively estimated basing on previously collected data and patient subjective clinical reports. This does not represent an objective clinical data and it cannot be used in the analysis. Therefore, only HY stage should be used as a motor impairment score. This point should be discussed as a significant limit of the study.

B) A multivariate logistic regression analysis considering PD as a dependent variable was performed to assess the role of possible confounders in the association between PD and higher PD levels. However, the Authors state “We found that the statistically significant correlation between LRG levels and the presence of PD was preserved after adjusting for CCI scores and CRP levels”. This sentence must be rephrased considering that it is an association and not a correlation.

C) Similarly, also the association between higher LRG levels and dementia should be adjusted by potential confounding variables (e.g age) through logistic regression analysis.

D) Multiple linear regression analysis was performed to confirm the positive correlation between serum LRG levels and HY stage in PD group. However, I believe it is better to consider LRG levels as the dependent variable.

Minor comments:

A) Table 3 could be avoided and adjusted p-values could be reported in table 1.

7. PLOS authors have the option to publish the peer review history of their article (what does this mean?). If published, this will include your full peer review and any attached files.

Reviewer #1: No

Reviewer #2: No

---

## [Author Response · Author response to Decision Letter 1]

3 Feb 2023

Dear Reviewers

We would like to thank for your time for evaluating our manuscript and also for your constructive comments that stimulated our manuscript to get much stronger. We have addressed all the points raised by the reviewers and our responses are described in the attached point by point response. 

Takashi Kasai

---

## [Editor Report · Decision Letter 2]

8 Feb 2023

Serum leucine-rich α2 glycoprotein as a potential biomarker for the systemic inflammation in Parkinson's disease.

PONE-D-22-21867R2

Dear Dr. Kasai

I appreciate the effort in improving your manuscript following reviewers' suggestions and we’re pleased to inform you that your manuscript has been judged scientifically suitable for publication and will be formally accepted for publication once it meets all outstanding technical requirements.

Kind regards,

Giulia Donzuso

Academic Editor

PLOS ONE

---

## [Editor Report · Acceptance letter]

13 Feb 2023

PONE-D-22-21867R2 

Serum leucine-rich α2 glycoprotein as a potential biomarker for systemic inflammation in Parkinson’s disease 

Dear Dr. Kasai:

I'm pleased to inform you that your manuscript has been deemed suitable for publication in PLOS ONE. Congratulations! Your manuscript is now with our production department. 

Kind regards, 

on behalf of

Dr. Giulia Donzuso 

Academic Editor

PLOS ONE